# Nutrition and Lifestyle Interventions in Managing Dyslipidemia and Cardiometabolic Risk

**DOI:** 10.3390/nu17050776

**Published:** 2025-02-23

**Authors:** Hygerta Berisha, Reham Hattab, Laura Comi, Claudia Giglione, Silvia Migliaccio, Paolo Magni

**Affiliations:** 1Dipartimento di Scienze Farmacologiche e Biomolecolari, Università degli Studi di Milano, 20133 Milan, Italy; hygerta.berisha@unimi.it (H.B.); reham.hattab@unimi.it (R.H.); laura.comi1@unimi.it (L.C.); claudia.giglione@unimi.it (C.G.); 2Dipartimento di Medicina Sperimentale, Sapienza Università di Roma, 00185 Roma, Italy; silvia.migliaccio@uniroma1.it; 3IRCCS MultiMedica, Sesto San Giovanni, 20099 Milan, Italy

**Keywords:** nutritional management, dyslipidemia, cardiovascular risk, metabolic risk, cholesterol, triglycerides, lifestyle interventions

## Abstract

Dyslipidemia, characterized by abnormal blood lipid levels, is a major public health concern due to its association with atherosclerotic cardiovascular disease (ASCVD) and other cardiometabolic disorders. In this context, appropriate nutrition patterns are pivotal as they represent the basic approach for providing a wide range of substantial advantages. The best evidence for dyslipidemia management is offered by the Mediterranean Diet, the Plant-Based Diet, the High-Fiber Diet and the Anti-inflammatory Diet, while the DASH Diet and the Ketogenic Diet have also been shown to target additional pathological features like hypertension and other comorbidities. The bioactive compounds that are enriched in these nutrition patterns and able to manage dyslipidemia include monounsaturated fatty acids such as ω-3, polyphenols such as oleuropein, resveratrol, flavonoids, and catechins, carotenoids, phytosterols and soluble and unsoluble fibers. Diets rich in these compounds can improve lipid profile by mitigating oxidative stress, reducing low-grade chronic inflammation, modulating macronutrient absorption and other mechanisms, thereby supporting cardiovascular health. Additionally, lifestyle interventions such as regular physical activity, weight loss, reduced alcohol consumption and smoking cessation further ameliorate lipid metabolism and manage circulated lipid profile. Furthermore, emerging insights from nutrigenomics underscore the potential for proper diet to address genetic factors and optimize treatment outcomes. The pivotal role of nutrition interventions in the context of dyslipidemia and its cardiometabolic implications is discussed in this review, emphasizing evidence-based and personalized approaches.

## 1. Introduction

The significant incidence of dyslipidemia, defined as abnormal lipid levels in the blood, represents a major public health concern and highlights the importance of developing and implementing effective management strategies [1]. Elevated levels of low-density lipoprotein cholesterol (LDL-C) reduced high-density lipoprotein cholesterol (HDL-C) and increased triglycerides (TG) contribute to the disruption of lipid homeostasis, which plays a central role in the development of atherosclerotic cardiovascular disease (ASCVD) and cardiometabolic disorders [2,3]. Specifically, dyslipidemia drives endothelial cell dysfunction, chronic low-grade inflammation, atherogenesis and insulin resistance, amplifying the risk of cardiovascular and metabolic diseases [4,5]. Moreover, recent studies have underscored the role of gut microbiota in dyslipidemia and cardiovascular risk. Nutrition and lifestyle interventions provide a pivotal approach, addressing biomarkers and related pathophysiological pathways which may be followed, when appropriate, by pharmacological lipid-lowering therapies targeting key markers like LDL-C and TG to reduce cardiovascular risk [6]. Diets rich in unsaturated fats, fiber, antioxidants and anti-inflammatory bioactive compounds have been shown to improve lipid profiles by raising HDL-C, lowering TG levels and mitigating oxidative stress [7]. Regular physical activity, weight management and smoking cessation further improve lipid metabolism and reduce inflammation [8,9]. Beyond lipid modulation, proper nutrition plays a crucial role in supporting a healthy gut microbiota, which in turn influences systemic metabolic health and cardiovascular outcomes [10]. Advances in nutrigenomics highlight the potential for proper dietary interventions to optimize treatment outcomes by accounting for genetic predispositions. This narrative review aims to analyze the role of nutrition and lifestyle patterns in modulating lipid and non-lipid biomarkers, evaluating their integration into current therapeutic guidelines. The review highlights the pathophysiological mechanisms underlying dyslipidemia and its cardiovascular and metabolic implications while exploring advancements in diagnostic, nutrigenomic and lipid profiling for effective management. A search was conducted in the PubMed database using key words such as “nutrition”, “dyslipidemia”, “atherosclerotic cardiovascular disease”, “lipid biomarkers”, “cardiometabolic risk”, “lifestyle strategies”, “bioactive compounds”, “gut microbiota”, “inflammation”, “insulin sensitivity”, “oxidative stress”, “pharmacological therapies” and “dietary fats” initially identifying approximately 15,000 records. To refine the selection, filters were applied for the publication period (2013–2025), full text availability and specific article types. After further screening the remaining articles based on relevant titles and abstracts, a total of 98 articles were selected for inclusion in this review.

## 2. Understanding Dyslipidemia: Causes, Phenotypes and Clinical Implications

Dyslipidemia pathophysiology is multifactorial, encompassing genetic, dietary, and lifestyle factors. Unhealthy dietary habits, such as high intake of saturated fats, trans fats, refined sugars and a lack of fiber, are major contributors to the development of dyslipidemia [11]. A diet rich in processed foods, coupled with excessive alcohol consumption and sedentary lifestyles, exacerbates lipid imbalances and increases the risk of conditions like atherosclerosis and metabolic syndrome [12]. These factors not only impact lipid profiles, but also disrupt insulin sensitivity, promote oxidative stress and induce inflammation, which are key processes that link dyslipidemia to cardiovascular and metabolic disorders [13]. Genetic predisposition plays a pivotal role as well, with mutations in genes such as LDLR, APOB and PCSK9 contributing to familial hypercholesterolemia (FH), a condition marked by elevated LDL-C levels [14,15]. Mixed dyslipidemia, a combination of high TG, high LDL-C and low HDL-C, is frequently observed in conditions like type 2 diabetes mellitus (T2DM) and metabolic syndrome [16]. Moreover, hypertriglyceridemia (HTG), defined by elevated triglyceride levels, poses additional risks, particularly when accompanied by low HDL-C [17]. These conditions are influenced by both genetic mutations and environmental factors such as poor dietary habits and physical inactivity, which exacerbate the risk of cardiovascular diseases and other metabolic disorders, including insulin resistance, T2DM and acute pancreatitis [18]. Atherogenic dyslipidemia features an imbalanced lipid profile, including elevated LDL-C and TG, along with reduced HDL-C levels. This condition is frequently worsened by lifestyle factors such as high-fat diets, obesity and insulin resistance, which collectively heighten the risk of developing cardiovascular diseases [19]. Indeed, diabetic dyslipidemia, associated with T2DM, is characterized by HTG, low HDL-C and increased small dense LDL-C, driven by insulin resistance and impaired lipid metabolism, elevating cardiovascular risk [20]. Additionally, dysbetalipoproteinemia is a genetic disorder caused by ApoE mutations, leading to remnant lipoprotein buildup, elevated lipids and increased atherosclerosis risk [21]. Another genetic disorder of dyslipidemia is hyperchylomicronemia which is marked by excessive chylomicron accumulation, resulting in very high TG levels, low cardiovascular risk but high risk for metabolic complications like acute pancreatitis [22]. The symptoms of dyslipidemia are often subtle or nonspecific but can include physical manifestations such as xanthomas that is fatty deposit under the skin or pancreatitis in severe cases of HTG [23,24]. These symptoms highlight the underlying metabolic disturbances that can lead to more severe cardiovascular complications, such as coronary artery disease, stroke and peripheral artery disease. The classification of dyslipidemia into specific types is crucial for identifying underlying genetic and metabolic disturbances and for tailoring effective therapeutic strategies. Understanding these classifications not only aids in early diagnosis but also informs targeted interventions, emphasizing the foundational role of nutrition and other lifestyle interventions in managing these conditions. Table 1 summarizes the classification of dyslipidemia phenotypes, highlighting their key biomarkers and associated cardiometabolic risks based on the latest guidelines discussed in this section.

## 3. Pathophysiological Links Between Dyslipidemia and Cardiometabolic Risk

Lipid metabolism is a dynamic and interconnected process that plays a pivotal role in maintaining cardiovascular and metabolic health. Central to this process are lipoproteins, which are lipid–protein complexes that transport hydrophobic lipids through the aqueous environment of the bloodstream [25]. HDL-C is essential in regulating LDL-C levels through reverse cholesterol transport. This process, primarily occurring in the liver and peripheral tissues, involves HDL collecting excess cholesterol from cells and returning it to the liver for elimination or reuse [26]. Impaired HDL-C function can result in elevated LDL-C levels in the bloodstream, heightening the risk of atherosclerosis and cardiovascular diseases, especially in dyslipidemic conditions like T2DM and metabolic syndrome [27]. Key transporters and apolipoproteins, such as apolipoprotein B (ApoB) and apolipoprotein A-I (ApoA-I), serve as critical biomarkers in lipid homeostasis. ApoB is primarily associated with atherogenic lipoproteins, including very low-density lipoprotein cholesterol (VLDL-C) and LDL-C, while ApoA-I is the main component of HDL-C, which is known for its protective cardiovascular effects [28]. The balance between apolipoprotein B (ApoB) and apolipoprotein A-I (ApoA-I) is crucial for cardiovascular health. Elevated ApoB levels increase cardiovascular risk due to their atherogenic potential, while higher ApoA-I levels are linked to reduced risk [29]. As the ApoB/ApoA-I ratio rises, the risk of cardiovascular disease also increases, highlighting the importance of maintaining this balance [30]. Lipoprotein lipase (LPL) is another critical enzyme that hydrolyzes TG in circulating lipoproteins, facilitating their uptake by tissues and thereby influencing lipid levels in the plasma [31]. Dysregulation of LPL activity can lead to atherogenic dyslipidemia, characterized by elevated TG and reduced HDL-C levels, which are hallmark features of metabolic syndrome and associated with increased cardiovascular risk [32]. Furthermore, apolipoprotein C-III (ApoC-III) has emerged as a significant regulator of lipid metabolism, inhibiting LPL activity and contributing to HTG, which further exacerbates cardiovascular risk [33]. Figure 1 illustrates the pathophysiological mechanisms and biomarkers linking dyslipidemia to cardiometabolic risks and is discussed in this section.

Indeed, the interplay between dyslipidemia and various metabolic disturbances is multifaceted, involving systemic inflammation, oxidative stress and insulin resistance, which collectively contribute to increased cardiometabolic risk [34]. Dyslipidemia can arise from a combination of dietary factors, lifestyle choices and genetic predispositions that disrupt normal lipid metabolism. Diets high in saturated fats and sugars have been shown to lead to dyslipidemia by increasing TG and decreasing HDL-C levels, thereby promoting an atherogenic lipid profile [35], which is often accompanied by elevated inflammatory markers such as C-reactive protein (CRP) and interleukin-6 (IL-6) [36]. The relationship between dyslipidemia and oxidative stress is also significant. Increased levels of reactive oxygen species (ROS) and lipid peroxidation products, such as malondialdehyde (MDA), are indicative of increased oxidative stress that can also lead to the oxidation of LDL-C [37]. This oxidative modification of LDL-C is a key event in the development of atherosclerosis, as oxidized LDL is more readily taken up by macrophages, leading to foam cell formation and plaque development in arterial walls [38]. Moreover, dyslipidemia is closely linked to insulin resistance, a hallmark of metabolic syndrome and T2DM. Insulin resistance promotes increased free fatty acid flux to the liver, resulting in the overproduction of TG and VLDL-C [39]. This dysregulation of lipid metabolism not only alters the lipid profile but also contributes to vascular dysfunction and metabolic disturbances, including hypertension, increased cardiovascular risk and metabolic dysfunction-associated steatotic liver disease (MASLD) [40,41]. Recent studies have also highlighted the role of gut microbiota in the pathophysiology of dyslipidemia. Gut dysbiosis, characterized by an imbalance in gut microbiota, contributes to the production of pro-atherogenic metabolites like trimethylamine-*N*-oxide (TMAO), exacerbating lipid imbalances and promoting atherosclerosis [42]. Notably, individual unhealthy dietary patterns significantly influence gut microbiota composition, linking nutrition to subclinical carotid atherosclerosis and systemic metabolic regulation risk [43,44]. Gut dysbiosis can be associated with increased intestinal permeability, which further triggers systemic inflammation and promotes atherogenesis. This suggests a complex interplay between gut health and cardiovascular outcomes, where alterations in gut microbiota composition may influence lipid metabolism and inflammatory processes [45]. Understanding impaired lipid metabolism and pathophysiological-related mechanisms not only enhances our comprehension of cardiometabolic risk factors but also opens new avenues for proper therapeutic strategies. The integration of nutrition and lifestyle interventions with pharmacological treatments may provide a comprehensive approach to the management of dyslipidemia and its associated cardiovascular risks.

## 4. Nutritional Management of Dyslipidemia and Cardiometabolic Risk Biomarkers

The management of dyslipidemia has become increasingly refined, with medical nutrition therapy (MNT) strategies tailored to the specific type of dyslipidemia and associated cardiovascular risk factors. MNT has been shown to improve metabolic profiles in patients with dyslipidemia, even in the absence of lipid-lowering medications [46,47]. Certainly, dyslipidemia is linked to cardiometabolic risk factors, where nutrition and lifestyle interventions play a key role in its management by improving lipid profiles and modulating relevant biomarkers [48]. The Mediterranean Diet has gained recognition as an effective intervention for managing dyslipidemia and related cardiovascular complications. Numerous studies indicate that adherence to the Mediterranean Diet is associated with significant reductions in total cholesterol (TC) and LDL-C levels, particularly beneficial for individuals with atherogenic dyslipidemia and FH [49,50]. This dietary pattern, rich in monounsaturated fatty acids (MUFAs) along with phytochemicals like polyphenols, plays a crucial role in lipid metabolism by reducing LDL oxidation, enhancing HDL-C function and mitigating inflammatory processes [51]. MUFAs from sources such as olive oil and nuts promote hepatic clearance of LDL-C particles from the bloodstream and enhance HDL-C structure, thereby improving cholesterol efflux capacity [52]. Polyphenols, including key compounds such as oleuropein, resveratrol, flavonoids and catechins, neutralize reactive oxygen species (ROS), prevent LDL oxidation and downregulate NF-kB signaling, resulting in decreased production of IL-6, TNF-α and CRP [53,54]. Recent findings also highlight the positive impact of polyphenols and omega-3 fatty acids in lowering ApoB levels by upregulating the LDL-C receptor, which is a critical indicator of cardiovascular risk, especially in subjects with T2DM [55,56]. The integration of this diet with pharmacological treatments, such as statins and other lipid modifying drugs, has been shown to amplify lipid-lowering effects, leading to lower cardiovascular risk [57]. Furthermore, the influence of geographical and social factors on dietary adherence underscores the necessity for tailored strategies to promote the diet in Mediterranean regions and beyond. The PREDIMED-Plus trial showed that adherence to a Mediterranean Diet supplemented with either nuts or extra virgin olive oil reduces hypertension and LDL-C levels, and improves insulin sensitivity and HDL levels, resulting in reduced cardiovascular events [58,59]. Continued research is necessary to elucidate the effects of this dietary pattern on the metabolome and microbiome across diverse populations, enhancing our understanding of its comprehensive health benefits [60]. Similarly, the Dietary Approaches to stop Hypertension (DASH) diet, which emphasizes fruits, vegetables, whole grains and low-fat dairy, effectively lowers LDL-C and TC levels, with a specific beneficial effect on diabetic dyslipidemia and mixed hyperlipidemia [61]. The phytochemicals in these foods, including flavonoids, carotenoids and phytosterols, further support cardiovascular health by boosting antioxidant activity, and improving endothelial function [62]. The ketogenic diet, characterized by low carbohydrate and high fat intake, has been shown to positively influence lipid profiles, particularly by lowering TG levels and elevating HDL-C [63]. This effect is attributed to the emphasis of the diet on healthy fats, such as those from olive oil and fatty fish, which provide essential fatty acids and antioxidants that contribute to these favorable lipid changes. Studies have indicated that the ketogenic diet can be effective for individuals with HTG, although it might be in some cases associated with significant increases in LDL-C levels [64,65]. Plant-based diets, rich in fiber and phytochemicals, have been shown to effectively lower TC and LDL-C levels, making them particularly suitable for individuals with dysbetalipoproteinemia and diabetic dyslipidemia [66,67]. Furthermore, plant-based diets can increase the levels of ApoA1 and decrease the levels of ApoB, thereby improving the ApoB/ApoA1 ratio [68]. The fiber-rich diets have been shown to positively influence metabolic markers, including insulin sensitivity and inflammatory markers, while also impacting the levels of various lipoproteins involved in lipid transport [69]. In patients receiving lipid-modifying therapy, such as statins or fibrates, high-fiber diets can further enhance lipid-lowering effects and improve cardiovascular risk profiles [70]. Recent clinical studies indicate that the consumption of water-soluble, viscous dietary fibers can significantly lower TC and LDL-C levels, with reductions ranging from 5% to 10%, while also highlighting the importance of fiber type, intake amount and dietary context in influencing these outcomes [71]. Diets rich in food with abundant anti-inflammatory components, such as polyphenols, ω-3 fatty acids, fiber and essential micronutrients, modulate lipid metabolism, reduce oxidative stress and improve dyslipidemia by targeting chronic inflammation [72]. The integration of dietary interventions and pharmacological treatment can effectively manage dyslipidemia across its various phenotypes, influencing lipid markers, apolipoprotein levels and cardiometabolic risk factors, since their combination can lead to enhanced therapeutic outcomes. Table 2 presents the main outcomes of studies on dyslipidemia phenotypes, associated dietary patterns and biomarker modulation discussed in this section.

## 5. Integration of Nutritional Management of Dyslipidemia with Lifestyle Approaches

The cardiometabolic advantages of dietary interventions in patients with dyslipidemia may be expanded by the implementation of lifestyle modifications [73]. Lipid profile is improved by regular physical activity, since exercise not only enhances lipid metabolism by increasing the activity of fat oxidation enzymes, but also improves insulin sensitivity, which is crucial for mitigating the risk of metabolic syndrome and cardiovascular diseases [74]. Research indicates that individuals engaging in regular physical activity experience lower TG and LDL-C levels while simultaneously increasing HDL-C profiles [75]. By increasing HDL-C and decreasing TG levels, with structured aerobic exercise programs, enhanced lipid-lowering effects are shown when combined with pharmacological therapies [76]. The combination of nutrition and physical exercise is pivotal for overweight management as even weight loss can lead to significant improvements in lipid profiles and metabolic markers, including reductions in LDL-C and TG [77]. Current evidence suggests that weight management not only reduces adipose tissue mass but also lowers inflammation by reducing plasma levels of CRP, leptin, IL-6 and TNF-α [78,79]. Numerous studies highlight the benefits of bariatric surgery beyond weight loss including improvements in lipid profile, obesity-related conditions and overall quality of life. Sleeve gastrectomy (SG) has been shown to significantly reduce TG and increase HDL-C in morbidly obese patients, while TC and LDL-C remained unchanged [80]. Despite study variations, it also mitigates subclinical atherosclerosis and improves endothelial function, contributing to reduced cardiovascular risk [81]. Longer follow-up studies are needed for stronger evidence. Smoking cessation is equally important, as smoking exacerbates dyslipidemia by increasing oxidative stress and inflammation, leading to a more atherogenic lipid profile; thus, cessation can reverse these effects and improve lipid levels [82,83]. Moderating alcohol consumption, improving sleep quality, managing stress and ensuring adequate hydration are also important components of lifestyle modifications that can positively influence lipid metabolism and overall cardiovascular health [84,85]. The combination of these interventions can lead to synergistic effects, resulting in improved health outcomes for individuals with dyslipidemia [86].

## 6. Precision Nutrition for Dyslipidemia: Future Perspectives

Precision nutrition promises to transform dietary advice by tailoring it to individual genetics and environment, but its implementation depends on continued research, as current evidence is scarce [87]. Nutrigenomics investigates how genetic variations affect dietary responses, particularly in lipid metabolism, and some examples are reported here below. Variations in the APOA5 gene have been shown to interact with dietary fat intake, significantly impacting obesity and TG levels in a Mediterranean population. The minor C allele of the APOA5 gene (rs662799) is negatively associated with the effect on TG, insulin levels and HOMA-IR after consuming a low-calorie Mediterranean Diet [88]. In the context of the PREDIMED Study, it was found that the rs1260326 variant of the glucokinase regulatory protein gene is significantly associated with higher TG concentrations which were improved by adherence to the Mediterranean Diet [89]. A clinical trial has provided strong evidence that omega-3 fatty acid supplementation significantly reduces LDL-C, TC and serum TG over three months in individuals with PPARG single nucleotide polymorphisms, who have elevated plasma cholesterol levels and are at low-to-moderate cardiovascular risk [90]. These findings suggest that genetic profiling can optimize dietary recommendations for lipid management and suggest some caution in proposing the very same nutritional strategy to all individuals, since the lipidemic profile changes may not be positive in all cases. As the field progresses, future directions may include leveraging digital health technologies and multiomic approaches to refine dyslipidemia management further [91]. This precision approach not only facilitates effective dyslipidemia management but also aids in preventing associated cardiovascular complications [92].

## 7. Conclusions

In conclusion, effective management of dyslipidemia requires a comprehensive approach that incorporates significant dietary and lifestyle interventions and, when appropriate, pharmacological therapies. In this review article, we discussed the current evidence regarding the benefits of selected dietary approaches for dyslipidemia modulation and improvement of cardiovascular and metabolic outcomes [93,94,95]. From a practical standpoint, the application of this knowledge is its actual implementation in the general population and in some specific subsets, which is a process that should take into consideration several additional aspects (geographical, cultural, religious, ethnic, etc.) to be effective [96]. As we move forward, the integration of proper approaches, guided by optimization of lifestyle interventions and possibly nutrigenomics, will be essential in a more effective management of dyslipidemia [97]. Additionally, advancements in artificial intelligence (AI) and machine learning (ML) offer great promise in revolutionizing diagnostics and treatment strategies for dyslipidemia and cardiometabolic diseases [98], paving the way for more personalized, efficient and impactful strategies in reducing cardiometabolic risk and improving long-term health outcomes.

## Figures and Tables

**Figure 1 nutrients-17-00776-f001:**
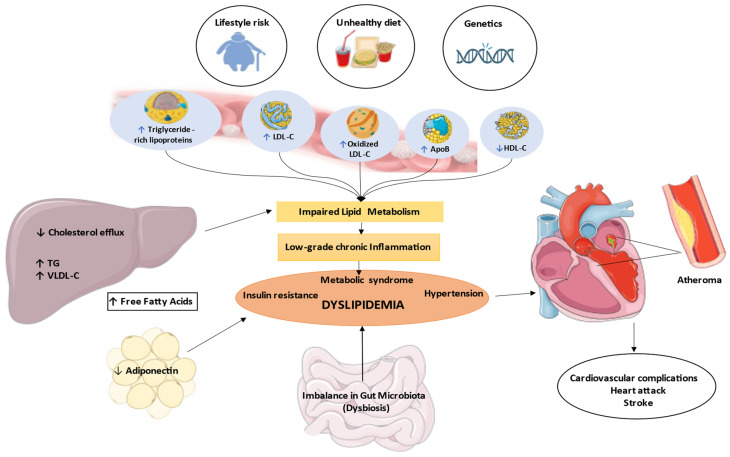
Dyslipidemia-related pathophysiological mechanisms of cardiometabolic disease risk. ↑: increased; ↓: reduced. The direction of the long arrows indicates the effect or impact on downstream organs.

**Table 1 nutrients-17-00776-t001:** Dyslipidemia phenotypes and cardiometabolic risk.

Phenotype	Key Biomarkers	Cardiovascular/Metabolic Risk	References
Atherogenic dyslipidemia	↑ TG, ↓ HDL-C, ↑ LDL	Moderate	[19]
Hypertriglyceridemia	Mild-to-moderate ↑ TG, ↓ HDL-C	Moderate to high	[17]
Diabetic dyslipidemia	↑ TG, ↓ HDL-C, ↑ LDL-C, hyperglycemia	High	[20]
Mixed dyslipidemia	↑ LDL-C, ↑ TG, ↓ HDL-C	High	[16]
Familial hypercholesterolemia	Marked ↑ LDL-C, genetic mutations (LDLR, APOB, PCSK9)	Very high	[14]
Dysbetalipoproteinemia (Type III)	↑ total cholesterol, genetic mutations (APOE)	Very high	[21]
Hyperchylomicronemia	Extremely ↑ TG, fasting chylomicrons	Low/high	[22]

↑: increased; ↓: reduced.

**Table 2 nutrients-17-00776-t002:** Nutrition intervention on lipid biomarkers and beyond.

Dietary Pattern	Dyslipidemia Phenotypes	Lipid Biomarkers	Cardiometabolic Markers	References
Mediterranean Diet	Familial hypercholesterolemia, atherogenic dyslipidemia	↓ TC, ↓ LDL-C, ↓ ApoB, ↑ HDL-C, ↓ LDL oxidation	↓ CRP, ↓ IL-6, ↓ TNF-α, ↓ ROS	[49,50,51,52,53,54,55,56,57]
PREDIMED Diet	Hypertriglyceridemia	↑ HDL-C, ↓ LDL-C, ↓ TG	↓ Hypertension, ↓ insulin sensitivity	[58,59]
DASH Diet	Diabetic dyslipidemia, Mixed hyperlipidemia	↓ LDL-C, ↓ TC,	↓ Hypertension, ↓ endothelial dysfunction	[61,62]
Ketogenic Diet	Hypertriglyceridemia	↓ TG, ↑ HDL-C	Improved insulin sensitivity, but risk of elevated LDL-C	[63,64,65]
Plant-Based Diet	Dysbetalipoproteinemia, Diabetic dyslipidemia	↓ TC, ↓ LDL-C, ↓ ApoB, ↑ ApoA1	↓ Inflammation, improved ApoB/ApoA1 ratio, ↑ antioxidant activity	[66,67,68]
High-Fiber Diet	Familial hypercholesterolemia, mixed dyslipidemia	↓ TC, ↓ LDL-C	↑ Insulin sensitivity, ↓ systemic inflammation	[69,70,71]
Anti-inflammatory Diet	Atherogenic dyslipidemia, mixed dyslipidemia	Modulation of lipid metabolism	↓ Oxidative stress, ↓ chronic inflammation, enhanced lipid regulation	[72]

↑: increased; ↓: reduced.

## Data Availability

Not applicable.

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
