# Peer review of "Nutrition and Lifestyle Interventions in Managing Dyslipidemia and Cardiometabolic Risk"

_nutrients, 2025, doi:10.3390/nu17050776_

Round 1

Reviewer 1 Report

Comments and Suggestions for Authors

Dear Authors,

The work is interesting and raises the importance of nutritional and lifestyle interventions in treating dyslipidemia and cardiometabolic risk.

However, in my opinion, the abstract is too general and should present the most important findings of this review.

The Introduction does not indicate how many of these articles were found in the databases by searching for the terms "nutrition," "dyslipidemia," "atherosclerotic cardiovascular disease," "lipid biomarkers," "cardiometabolic risk," "lifestyle strategies," "bioactive compounds," "gut microbiota," "inflammation," "insulin sensitivity," "oxidative stress," "pharmacological therapies," and “dietary fats.” How were the articles selected for this article, and whether there are only these in the database, or were further selections made?

Figure 1 is like a graphical abstract.

The conclusions of this review are a bit disappointing because they don't add much beyond the general knowledge of those interested in or involved in nutrition and dietetics.

The references are cited contrary to the journal Nutrients rules, which should be corrected.

In my opinion, the article should be further polished because it can become a well-written manuscript.

Reviewer

Author Response

Reply to Reviewer 1

The work is interesting and raises the importance of nutritional and lifestyle interventions in treating dyslipidemia and cardiometabolic risk.

However, in my opinion, the abstract is too general and should present the most important findings of this review.

Thank you for your valuable feedback. We have revised the abstract to highlight the key findings of our review more clearly and concisely, as follows:

Abstract: Dyslipidemia, characterized by abnormal blood lipid levels, is a major public health concern due to its association with atherosclerotic cardiovascular disease (ASCVD) and other cardiometabolic disorders. In this context, appropriate nutrition patterns are pivotal as they represent the basic approach for providing a wide range of substantial advantages. The best evidence for dyslipidemia management is offered by the Mediterranean Diet, the Plant-Based Diet, the High-Fiber Diet and the Anti-inflammatory Diet, while the DASH Diet and the Ketogenic Diet have been also shown to target additional pathological features like hypertension and other comorbidities. The bioactive compounds that are enriched in these nutrition patterns and able to manage dyslipidemia include monounsaturated fatty acids such as ω-3, polyphenols such as oleuropein, resveratrol, flavonoids, and catechins, carotenoids, phytosterols and soluble and unsoluble fibers. Diets rich in these compounds can improve lipid profile by mitigating oxidative stress, reducing low-grade chronic inflammation, modulating macronutrient absorption and other mechanisms, thereby supporting cardiovascular health. Additionally, lifestyle interventions such as regular physical activity, weight loss, reduced alcohol consumption and smoking cessation further ameliorate lipid metabolism and manage circulated lipid profile. Furthermore, emerging insights from nutrigenomics underscore the potential for proper diet to address genetic factors and optimize treatment outcomes. The pivotal role of nutrition interventions in the context of dyslipidemia and its cardiometabolic implications is discussed in this review emphasizing evidence-based and personalized approaches.   

The Introduction does not indicate how many of these articles were found in the databases by searching for the terms "nutrition," "dyslipidemia," "atherosclerotic cardiovascular disease," "lipid biomarkers," "cardiometabolic risk," "lifestyle strategies," "bioactive compounds," "gut microbiota," "inflammation," "insulin sensitivity," "oxidative stress," "pharmacological therapies," and “dietary fats.”

How were the articles selected for this article, and whether there are only these in the database, or were further selections made?

Thank you for this important suggestion. We revised the “methodology” paragraph as requested, as follows:

A search was conducted in the PubMed database using key words such as “nutrition”, “dyslipidemia”, “atherosclerotic cardiovascular disease”, “lipid biomarkers”, “cardiometabolic risk”, “lifestyle strategies”, “bioactive compounds”, “gut microbiota”, “inflammation”, “insulin sensitivity”, “oxidative stress”, “pharmacological therapies” and “dietary fats” initially identifying approximately 15,000 records. To refine the selection, filters were applied for the publication period (2013–2025), full text availability, and specific article types. After further screening the remaining articles based on relevant titles and abstracts, a total of 98 articles were selected for inclusion in this review.

Figure 1 is like a graphical abstract.

Thank you for your feedback. Figure 1 is intended to illustrate only the pathophysiological mechanisms and biomarkers linking dyslipidemia to cardiometabolic risk. We have revised Figure 1 to better summarize these pathophysiological links. We used a table (Table 2) to summarize the main focus of this review “the current evidence regarding specific nutrition approaches to manage dyslipidemia”.  

The conclusions of this review are a bit disappointing because they don't add much beyond the general knowledge of those interested in or involved in nutrition and dietetics.

We appreciate your perspective and have worked to refine the conclusions to better highlight the novel insights and key takeaways from our review.

In conclusion, effective management of dyslipidemia requires a comprehensive approach that incorporates significant dietary and lifestyle interventions and, when appropriate, pharmacological therapies. In this review article, we discussed the current evidence regarding the benefits of selected dietary approaches for dyslipidemia modulation and improvement of cardiovascular and metabolic outcomes [93–95]. From a practical standpoint, the application of this knowledge is its actual implementation in the general population and in some specific subsets, which is a process that should take into consideration several additional aspects (geographical, cultural, religious, ethnic, etc.) to be effective [96]. As we move forward, the integration of proper approaches, guided by optimization of lifestyle interventions and possibly nutrigenomics, will be essential in a more effective management of dyslipidemia [97]. Additionally, advancements in artificial intelligence (AI) and machine learning (ML) offer great promise in revolutionizing diagnostics and treatment strategies for dyslipidemia and cardiometabolic diseases [98], paving the way for more personalized, efficient and impactful strategies in reducing cardiometabolic risk and improving long-term health outcomes. 

The references are cited contrary to the journal Nutrients rules, which should be corrected.

Thanks for this comment, we took care of the references and corrected them when not aligned with Nutrients’ rules.

In my opinion, the article should be further polished because it can become a well-written manuscript.

Thanks, we further polished the text as requested.

Reviewer 2 Report

Comments and Suggestions for Authors

Dear Editor,

I'm pleased to have the opportunity to review the article entitled:

Nutrition and Lifestyle Interventions in Managing Dyslipidemia and Cardiometabolic Risk.

This review provides a comprehensive overview of dyslipidemia and its implications for cardiovascular health, effectively highlighting the interplay between nutritional and lifestyle factors alongside pharmacological interventions. The characterization of dyslipidemia as a major public health concern is well justified, particularly given its strong association with atherosclerotic cardiovascular disease (ASCVD) and other metabolic disorders.

The discussion on the specific lipid profiles—elevated LDL cholesterol, reduced HDL cholesterol, and high triglycerides—nicely establishes the foundational knowledge needed to understand the pathophysiology at play, including insights into endothelial dysfunction, systemic inflammation, and metabolic issues.

Further, the review rightly emphasizes the critical role of nutrition and lifestyle modifications. The recommendations for diets rich in unsaturated fats, fiber, antioxidants, and anti-inflammatory components are well supported by contemporary research, showcasing their ability to enhance lipid profiles and mitigate oxidative stress. Additionally, the acknowledgment of regular physical activity, weight loss, and smoking cessation as effective methods to improve lipid metabolism and insulin sensitivity is a strong point, as these lifestyle changes have profound systemic benefits.

The mention of nutrigenomics adds an intriguing dimension, suggesting that tailored dietary interventions could help address individual genetic predispositions and enhance treatment outcomes. This personalized approach underscores the potential for optimizing management strategies in dyslipidemia, moving beyond a one-size-fits-all model.

Overall, this review successfully emphasizes evidence-based strategies and the importance of an individualized approach in managing dyslipidemia, making it a valuable resource for clinicians and researchers alike. Future investigations could build on these findings to explore more about the biological mechanisms linking lifestyle changes to improved lipid profiles and cardiovascular health.

In my opinion, the author should emphasize bariatric surgery's impact on weight reduction and comorbidities.

Author Response

This review provides a comprehensive overview of dyslipidemia and its implications for cardiovascular health, effectively highlighting the interplay between nutritional and lifestyle factors alongside pharmacological interventions. The characterization of dyslipidemia as a major public health concern is well justified, particularly given its strong association with atherosclerotic cardiovascular disease (ASCVD) and other metabolic disorders.

The discussion on the specific lipid profiles—elevated LDL cholesterol, reduced HDL cholesterol, and high triglycerides—nicely establishes the foundational knowledge needed to understand the pathophysiology at play, including insights into endothelial dysfunction, systemic inflammation, and metabolic issues.

Further, the review rightly emphasizes the critical role of nutrition and lifestyle modifications. The recommendations for diets rich in unsaturated fats, fiber, antioxidants, and anti-inflammatory components are well supported by contemporary research, showcasing their ability to enhance lipid profiles and mitigate oxidative stress. Additionally, the acknowledgment of regular physical activity, weight loss, and smoking cessation as effective methods to improve lipid metabolism and insulin sensitivity is a strong point, as these lifestyle changes have profound systemic benefits.

The mention of nutrigenomics adds an intriguing dimension, suggesting that tailored dietary interventions could help address individual genetic predispositions and enhance treatment outcomes. This personalized approach underscores the potential for optimizing management strategies in dyslipidemia, moving beyond a one-size-fits-all model.

Overall, this review successfully emphasizes evidence-based strategies and the importance of an individualized approach in managing dyslipidemia, making it a valuable resource for clinicians and researchers alike. Future investigations could build on these findings to explore more about the biological mechanisms linking lifestyle changes to improved lipid profiles and cardiovascular health.

Thank you very much for your positive comments on the structure and content of this review article.

In my opinion, the author should emphasize bariatric surgery's impact on weight reduction and comorbidities.

Thanks for this suggestion. Accordingly we added the following text in the appropriate section.

Numerous studies highlight the benefits of bariatric surgery beyond weight loss including improvements in lipid profile, obesity-related conditions, and overall quality of life. Sleeve gastrectomy (SG) has been shown to significantly reduce TG and increase HDL-C in morbidly obese patients, while TC and LDL-C remained unchanged. [80]. Despite study variations, it also mitigates subclinical atherosclerosis and improves endothelial function, contributing to reduced cardiovascular risk [81]. Longer follow-up studies are needed for stronger evidence.

Reviewer 3 Report

Comments and Suggestions for Authors

The authors have produced a detailed narrative review of the way in which nutritional and lifestyle interventions may help in the management of both dyslipidaemia and cardiovascular or metabolic risk factors. The review is comprehensive. The authors have included a good illustration.

A formal structured review would, of course, have been much better. However, the present work is easy to read and includes a large number of up-to-date references. There are no obvious errors of omission. The authors handle certain controversial issues, such as the relative benefits of different types of diets, in a fair and well-balanced manner.

Author Response

The authors have produced a detailed narrative review of the way in which nutritional and lifestyle interventions may help in the management of both dyslipidemia and cardiovascular or metabolic risk factors. The review is comprehensive. The authors have included a good illustration.

Thanks for your positive evaluation of this review article.

A formal structured review would, of course, have been much better. However, the present work is easy to read and includes a large number of up-to-date references. There are no obvious errors of omission. The authors handle certain controversial issues, such as the relative benefits of different types of diets, in a fair and well-balanced manner.

Thank you for your positive consideration.